# Analyzes In Silico Indicate the lncRNAs MIR31HG and LINC00939 as Possible Epigenetic Inhibitors of the Osteogenic Differentiation in PDLCs

**DOI:** 10.3390/genes14081649

**Published:** 2023-08-18

**Authors:** Rogério S. Ferreira, Rahyza I. F. Assis, Francesca Racca, Ana Carolina Bontempi, Rodrigo A. da Silva, Malgorzata Wiench, Denise C. Andia

**Affiliations:** 1School of Dentistry, Health Science Institute, Paulista University, São Paulo 04026-002, SP, Brazil; rogerio.ferreira25@aluno.unip.br (R.S.F.); anacarolina.bontempi@gmail.com (A.C.B.); 2Department of Clinical Dentistry, Federal University of Espírito Santo, Vitória 29043-910, ES, Brazil; 3Periodontology Department, The Ohio State University College of Dentistry, Columbus, OH 43210-1267, USA; francesca_racca@hotmail.com; 4Program in Environmental and Experimental Pathology, Paulista University, São Paulo 04026-002, SP, Brazil; dasilva.rodrigo.a@gmail.com; 5School of Dentistry, Institute of Clinical Sciences, Institute of Cancer and Genomic Sciences, University of Birmingham, Birmingham B5 7EG, UK

**Keywords:** PDLC, epigenetic, osteogenic, SP7, DLX4, MIR31HG, LINC00939

## Abstract

Chromatin conformation, DNA methylation pattern, transcriptional profile, and non-coding RNAs (ncRNAs) interactions constitute an epigenetic pattern that influences the cellular phenotypic commitment and impacts the clinical outcomes in regenerative therapies. Here, we investigated the epigenetic landscape of the SP7 transcriptor factor (*SP7*) and Distal-Less Homeobox 4 (*DLX4*) osteoblastic transcription factors (TFs), in human periodontal ligament mesenchymal cells (PDLCs) with low (l-PDLCs) and high (h-PDLCs) osteogenic potential. Chromatin accessibility (ATAC-seq), genome DNA methylation (Methylome), and RNA sequencing (RNA-seq) assays were performed in l- and h-PDLCs, cultured at 10 days in non-induced (DMEM) and osteogenic (OM) medium in vitro. Data were processed in *HOMER*, *Genome Studio,* and *edgeR* programs, and metadata was analyzed by online bioinformatics tools and in *R* and *Python* environments. ATAC-seq analyses showed the TFs genomic regions are more accessible in l-PDLCs than in h-PDLCs. In Methylome analyses, the TFs presented similar average methylation intensities (AMIs), without differently methylated probes (DMPs) between l- and h-PDLCs; in addition, there were no differences in the expression profiles of TFs signaling pathways. Interestingly, we identified the long non-coding RNAs (lncRNAs), *MIR31HG* and *LINC00939,* as upregulated in l-PDLCs, in both DMEM and OM. In the following analysis, the web-based prediction tool *LncRRIsearch* predicted RNA:RNA base-pairing interactions between *SP7*, *DLX4*, *MIR31HG,* and *LINC00939* transcripts. The machine learning program *TriplexFPP* predicted DNA:RNA triplex-forming potential for the *SP7* DNA site and for one of the LINC00939 transcripts (*ENST00000502479*). PCR data confirmed the upregulation of *MIR31HG* and *LINC00939* transcripts in l-PDLCs (× h-PDLCs) in both DMEM and OM (*p* < 0.05); conversely, *SP7* and *DLX4* were downregulated, confirming those results observed in the RNA-Seq analysis. Together, these results indicate the lncRNAs MIR31HG and LINC00939 as possible epigenetic inhibitors of the osteogenic differentiation in PDLCs by (post)transcriptional and translational repression of the *SP7* and *DLX4* TFs.

## 1. Introduction

Over the past decades, stem cell-based treatment associated with regenerative therapies has become increasingly promising for the treatment of several diseases such as diabetes, cardiac ischemia, and osteoarthritis [1]. Mesenchymal stem cells (MSCs) can be obtained from several sources, such as bone marrow (BMSCs), adipose tissue (ASCs), peripheral blood (HSCs) [2], and teeth (periodontal ligament cells—PDLCs, dental pulp cells—DPCs, and stem cell from exfoliated human dentition—SHED) [3]. MSCs are characterized by self-renewing, differentiation into cell multilineages capacities, and specific surface markers [4]. Moreover, MSCs show inherent tropism toward damaged tissues and the ability to regenerate them [5,6].

PDLCs can differentiate into osteoblastic, adipocyte, neuronal, and chondrogenic-like cells [7], although they might present distinct capacities to produce mineral nodules in vitro [8,9,10,11,12], which could impact clinical applications. This heterogeneity may be related to cell fate commitment, which is “the commitment of cells to specific cell fates and their capacity to differentiate into particular kinds of cells” (Gene_Ontology_Term_Definition_GO:0045165).

The cell fate commitment is determined by lineage-specific transcription factors (TFs), which are proteins that bind in DNA sites and drive the cellular phenotype acquisition [13]. These TFs are expressed through signaling pathways, which are cascades of extra and intracellular molecular events that culminate in the TFs gene expression. This entire molecular process is regulated by epigenetic mechanisms such as chromatin conformation, DNA methylation marks, and non-coding RNA (ncRNAs) interferences. Chromatin can remodel into a more condensed (heterochromatin) or less condensed (euchromatin) structure, determining the degree of RNA polymerase accessibility, responsible for gene transcription, at the DNA strand [14]. This conformation is modulated by epigenetic modifications in histones and DNA methylation patterns. The DNA methylation patterns, in turn, are determined by the intensity of the methyl group aggregation at carbon 5 of cytosine (5 mC) [15], favoring or not binding TFs in gene promoter regions [16]. ncRNAs are RNA molecules that are not translated into protein and can be distinguished according to their size: microRNAs (miRNAs), about 22 nucleotides long, and long non-coding RNAs (lncRNAs), over 200 nucleotides long. Mostly, the miRNAs act through RNA-induced silencing complex (RISC) as post-transcriptional silencers, promoting mRNA degradation by the cleavage mechanism or inhibiting its translation by the base pairing mechanism [17]. The lncRNAs can act as transcriptional regulators by the DNA:RNA triple-helix (triplex) forming, through Hoogsteen or reverse Hoogsteen base pairing, i.e., when a polypurine (A–G) or pyrimidine (C–U) motif of an lncRNA interacts with the major groove of a Watson–Crick double-stranded DNA (dsDNA), forming triplets of canonical bases, such as C•G–C and U•A–T (where ‘•’ and ‘–’ represent Hoogsteen and Watson–Crick interactions, respectively) [18,19]. In addition, they can also act as post-transcriptional and translational regulators by the RNA:RNA duplex intermolecular hybridization through nucleotide base pairing interactions between a lncRNA and an mRNA [20], among other mechanisms. However, such mechanisms remain poorly understood.

Previous studies from our group have demonstrated the individual epigenetic profile influences the capacity of extracellular matrix deposition, and, consequently, the osteogenic phenotype acquisition [9,11]; in addition, we also highlighted the *Sp7 Transcription Factor* (*SP7*) and *Distal-Less Homeobox 4* (*DLX4*) genes, key TFs involved in osteoblastic differentiation, as downregulated in PDLCs with a low capacity of mineral matrix deposition [12]. Here, our aim is to investigate, in silico, the epigenetic landscape of both TFs, *SP7,* and *DLX4*, in PDLCs showing the distinct capacity of mineral matrix deposition in vitro.

## 2. Material and Methods

### 2.1. Cell Acquisition and Culture

After signing an informed consent approved by the Ethics Committee of Piracicaba Dental School, University of Campinas (CAAE55588816.4.0000.5418), PDLCs were collected, isolated, and cultured as described by Silverio et al., 2010 [21]. Then, PDLCs were characterized into low (l-PDLCs) and high osteogenic potential (h-PDLCs), according to their capacity of mineral deposition, in vitro, and according to our previous publications [9,10,11,12]. Briefly, PDLCs were characterized according to Dominici et al. [4] to confirm the ability to differentiate into osteogenic and adipogenic cell lineages and the expression/lack of expression of specific cell surface markers, such as CD166, CD34, and CD45 [22]. The levels of CD34 and CD45 were very similar between h- and l-PDLCs, showing less than 1% of the expression of positive cells. Regarding multipotency marker CD166, more than 95% of cells in both populations showed positive expressions [9,10,11,12]. Alizarin red staining was performed to assess the amount of mineral matrix produced in vitro by each cell population [10]. Consequently, PDLCs were classified either as high osteogenic potential PDLCs (h-PDLCs), which was the cell population with the capacity to produce higher amounts of the mineral matrix or low osteogenic potential PDLCs (l-PDLCs with a lower capacity to produce mineral matrix). Based on our previous studies [10,12], we chose day 10 of the osteogenic media (OM) induction as the time point to analyze epigenomic and transcriptomic changes. l- and h-PDLCs were cultured in high glucose Dulbecco’s Modified Eagle Medium (DMEM), 10% Fetal Bovine Serum (FBS), 100 U/mL of penicillin, and 100 mg/mL of streptomycin and maintained at 37 °C in a humidified atmosphere containing 5% CO_2_. Three independent experiments were performed for each PDLC, with three technical replicates for each one, with cells in the passages P5-P8, except when stated otherwise.

### 2.2. Osteogenic Stimulation

Both l- and h-PDLCs were seeded into 6-well plates (2.5 × 10^5^ cells/well) in DMEM, 10% FBS, and antibiotics. After 24 h of incubation, for cell adhesion, the culture medium was removed, and the cells were cultivated in non-induced medium (DMEM) supplemented with 10% FBS and 1% antibiotics (penicillin 100 U/mL and streptomycin 100 mg/mL) or in induced osteogenic medium (OM) with DMEM, 10% FBS, 1% antibiotics, and supplemented with ascorbic acid (50 μg/mL), β-glycerophosphate (10 mM), and dexamethasone (10 nM). Cells were incubated and collected after 10 days, with media change every three days. PDLCs were divided according to the following groups:
(i)l-DMEM: l-PDLCs cultivated in DMEM, standard medium.(ii)l-OM: l-PDLCs cultivated in OM, osteogenic medium.(iii)h-DMEM: h-PDLCs cultivated in DMEM, standard medium.(iv)h-OM: h-PDLCs cultivated in OM, osteogenic medium.

For all comparisons, the h-DMEM was set as the control group x l-DMEM, and the h-OM was set as the control group x l-OM.

### 2.3. Assay for Transposase-Accessible Chromatin Using Sequencing (ATAC-Seq)

A total of 5 × 10^4^ cells were harvested from each group and were incubated in a transposition reaction, as preconized by Buenrostro et al. [23]. The Tn5 transposase enzyme was used to insert an adapter sequence into the accessible chromatin regions, combined with single-step library digestion and preparation. Digitonin was included to reduce contamination with mitochondrial DNA [24]. Then, the transposase-containing DNA fragments were amplified by PCR and purified to select the appropriate size of the fragments. Sample quality was evaluated by TapeStation, quantified by PCR with the Kapa Sybr Fast LightCycler 480 kit, and pooled for subsequent sequencing at Illumina NextSeq 500 platform (Illumina Inc., Foster City, CA, USA) in the Genomics Birmingham Facility (Birmingham, UK). Two independent experiments were performed.

### 2.4. Global DNA Methylation Analysis (Methylome)

#### DNA Isolation and Oxidative Bisulfite Conversion

The groups were cultured (8.7 × 10^5^ cells/100 mm dishes) as described above. After 10 days, the culture medium was removed, and the cells were washed two times with PBS and scrapped off in extraction buffer with proteinase K. Total DNA was purified by extraction with phenol/chloroform/isoamyl alcohol and stored at −20 °C. DNA’s concentrations and quality were assessed using Qubit (Thermo Fisher Scientific Inc., Rockford, IL, USA) and spectrophotometer (Nanodrop 1000; Nanodrop Technologies LLC, Wilmington, NC, USA). The oxidative bisulfite conversion reaction was performed according to the protocol described by Assis et al. [12].

### 2.5. RNA Sequencing (RNA-SEQ)

#### 2.5.1. I. RNA Extraction

Cells from all groups were cultivated at 1.5 × 10^5^ per well in 6-well plate, as previously described. After 10 days, the culture medium of each well from each group was removed, the cells were washed with PBS and scrapped off with TRIzol reagent (Invitrogen, Cat #15596-018, Waltham, MA, USA), according to the manufacturer’s recommendation. Total RNA extraction was performed, and RNA samples were treated with DNA-free Turbo solution to remove genomic DNA (Ambion, Cat #1907, Austin, TX, USA). Then, the samples were submitted to integrity and concentration analysis by the Agilent 2100 bioanalyzer, with an RNA Integrity Number (RIN) value greater than 8. 

#### 2.5.2. II. RNA Sequencing

The samples were pooled in equal concentration, prepared, and sequenced with Illumina TruSeq Stranded mRNA Sample Prep Kit, according to the manufacturer’s instructions, in Illumina NextSeq 500 platform (Illumina Inc., Foster City, CA, USA). Briefly, 1 μg of DNA-free total RNA samples were processed. The mRNA was fragmented and copied into the first strand cDNA, followed by second strand cDNA synthesis. cDNA fragments were submitted to the final repair process, addition of single adenosine base, and adapter ligation. Finally, the processed cDNA was amplified by 15 cycles of PCR to create the cDNA library, which was read on the HiSeq 2500 (v3) sequencer (Illumina, San Diego, CA, USA).

For more details about these genome wide methodologies, please go to Assis et al., 2022.

### 2.6. Bioinformatics Analysis

#### 2.6.1. III. ATAC-Seq

The reads were aligned to the human genome (h19) through the *Bowtie2 tool* [25] and the duplicates were removed. The ENCODE consortium identified the blacklist reads, defined as anomalous. Non-exclusively mapped reads have been filtered and peaks have been called using the “factor mode” in HOMER [26], based on the default settings. A Bigwig file was also generated and uploaded in the UCSC Genome Navigator [22] to allow visualization of the accumulated reads. Analyses using HOMER and subsequent analyses were conducted with the help of Dr. Samuel Clokie (West Midlands Regional Genetics Laboratory, Birmingham Women’s Hospital, Birmingham, UK).

#### 2.6.2. IV. DNA Methylome

Epic BeadChips data was processed through Illumina Genome Studio program [27] and minfi [28], implemented in R, associated with dplyr [29] and tidyr [30] packages. Data normalization was performed using quartiles methods. Probes were considered differentially methylated (DMP) when presented values of delta β > 0.2 (hypermethylated) or < −0.2 (hypomethylated) and p-value < 0.01. The average methylation intensity (AMI) of the TFs was calculated by the sum of the average β (AVG-β) values of the detected probes (sd), divided by the number of probes:(1)$$\bar{\mathrm{AMI}}=\frac{\sum AVG\_\beta \left(sd\right)}{n\left(sd\right)}.$$

The scatter plots representing the annotation, AMI, and AVG-β values and AMI of the TFs probes were generated in Microsoft Excel 365.

#### 2.6.3. RNA-Seq

The reads were aligned to the hg19 reference genome and counted using the R *Rsubread* package [31]. Quantification was performed according to the last recommended pipeline, as defined in the *edgeR* software [32]. Genes were considered differentially expressed (DEGs) when presented values of log2FoldChange (logFC) ≥ 1.5 (upregulated) or ≤−1.5 (downregulated), and *p*-value ≤ 0.05. “Heatmaps” representing the logFC intensity were generated in *R* environment with the ComplexHeatmap package [33], and the “volcano plots” representing both logFC amplitude and statistical significance (-log10pvalue), with the EnhancedVolcano package [34].

#### 2.6.4. Selection of lncRNAs and Prediction Analyses

The lncRNAs found DEGs (DElncRNAs) were upregulated in common between the RNA-seq datasets of the l-DMEM (× h-DMEM) and l-OM (× h-OM) groups and were selected and submitted to prediction analysis for RNA:RNA base-pairing interactions with the TFs, using the RIblast prediction program, based on the LncRRIsearch web server [35]. Since the TruSeq Stranded mRNA Sample Prep Kit, used in RNA Sequencing, is not quite suitable for non-coding RNA without polyA tails, we check if the selected DElncRNAs have polyadenylation sites from 3′ end sequencing, in the PolyASite database [36]. Next, they verified the potentials of DNA:RNA triplex formation of these predicted DElncRNAs and of the TFs, using the TriplexFPP machine learning program, in the Python environment, which (1) predicted the probabilities of triplex-forming oligonucleotide (TFO), in practice, for these DElncRNAs and (2) the potentials triplex target sites (TTSs) on the TFs DNA sequences, based on experimentally verified data, considering positive triplex-forming for score > 0.5 [37]. The FASTA sequences input in the program were obtained in the Ensembl genome browser, under the “Human (GRCh37.p13)” parameter [38]. The pie charts and Venn diagrams representing the distribution of the DElncRNAs among the groups was generated in Microsoft Excel 365 and with the online tool Interactivenn, respectively [39].

### 2.7. PCR Analysis

cDNA synthesis was performed using 1 mg RNA as described previously [40]. Quantitative polymerase chain reaction (PCR) was carried out for each one of the three independent experiments, using LightCycler 96 Real-Time PCR System (Roche Diagnostics GmbH, Mannheim, Germany) and FastStart Essential DNA Green Master (Roche Diagnostic Co., Indianapolis, IN, USA), according to the manufacturer’s instructions and in technical triplicates. The primers’ sequences and reaction details are shown in Appendix A. The results of *MIR31HG*, *LINC00939*, *SP7* and *DLX4* were obtained from three biological replicates, in technical triplicates, analyzed by ΔΔCt method [41] and are presented as relative amounts of the target gene using *18S* as inner reference gene.

### 2.8. Statistical Analysis of the PCR Data

Data were initially examined for normality by Shapiro–Wilk test and expressed as mean ± standard deviation (SD). After normal data distribution was confirmed, one-way analysis of variance (ANOVA α ≤ 0.05) followed by pairwise multiple-comparison test (Tukey) were used to identify the difference amongst groups (GraphPad Prism 7—GraphPad Software Inc., San Diego, CA, USA).

## 3. Results

### 3.1. l-PDLCs Show More Accessible Chromatin Regions in SP7 and DLX4 Genes than h-PDLCs

The chromatin conformation coordinates the DNA accessibility of the transcriptional machinery, composed of the RNA polymerase II, TFs, among other molecular elements, responsible for gene expression. The chromatin accessibility analysis on *SP7* and *DLX4* genomic regions in both h- and l-PDLCs, induced (h-OM and l-OM) and non-induced (h-DMEM and l-DMEM), to osteogenesis in vitro at 10 days showed more accessible chromatin peaks in l-PDLCs compared to h-PDLCs, in both culture conditions, i.e., induced, and non-induced (Figure 1). In summary, l-PDLCs exhibit chromatin conformation more favorable to *SP7* and *DLX4* genes transcription than h-PDLCs.

### 3.2. DNA Methylation Patterns amongst l- and h-PDLCS Are Similar for SP7 and DLX4

In turn, DNA methylation is one of the main epigenetic mechanisms that regulate chromatin conformation and gene transcription. We investigated the DNA methylation patterns of the *SP7* and *DLX4* genes in the Methylome metadata. Despite the probes presenting different AVG-β values in *SP7* and *DLX4*, the average methylation intensities (AMI) were similar, exhibiting a less methylated pattern, except for the 3′UTR region in *DLX4*, whose AMI pattern was more methylated (Figure 2). No differentially methylated probes (DMPs) were found between l- and h-PDLCs, in both culture conditions. These results reveal similar DNA methylation patterns among *SP7* and *DLX4* (except in the 3′UTR region), without significant differences between l- and h-PDLCs. There were no other found osteogenic TFs or signaling pathways that were differently expressed among l- and h-PDLCs.

The gene transcription mechanism is the biological consequence of molecular events that occur in extra and intracellular cascades called signaling pathways. Due to the inconsistency observed between the transcriptional profile and the epigenetic background of the TFs *SP7* and *DLX4* in our previous research [12], we analyzed the transcriptional profiles of the main signaling pathways involved in osteogenesis, in the l-DMEM (x h-DMEM), and in the l-OM (x h-OM) RNA-seq dataset. In addition, we analyzed the transcriptional profiles of other osteogenic TFs, like *RUNX2*, *SATB2*, *ATF4*, *MSX2*, *CTNNB1*, *DLX5,* and *TWIST1/2*. No differently expressed genes (DEGs) were found for these TFs (Figure 3A) or signaling pathways, except for *Sclerostin* (*SOST*), a negative regulator of the WNT/β-catenin pathway, founded DEG (downregulated) in both l-PDLCs groups (Figure 3B). This result points to no significant differences in the transcriptional profiles of other osteogenic TFs or signaling pathways between l- and h-PDLCs, in both culture conditions.

### 3.3. The lncRNAs LINC00939 and MIR31HG Are Upregulated in l-PDLCs

The lncRNAs are important agents of the epigenetic machinery that act on the transcriptional, post-transcriptional, and translational regulations. Therefore, we also investigated the lncRNAs transcriptional profiles. Of all 1643 lncRNAs identified, 73 (≈4.5%) were DEGs (DElncRNAs) in the l-DMEM group and 82 (≈5%) in the l-OM group. There were 18 DElncRNAs in common among the groups, of which four long intergenic non-protein coding RNAs (*LINCs 00939*, *01260*, *01347* and *02210*), four uncharacterized (*LOCs 286178*, *408186*, *100506476* and *100507547*), and one host-gene (*MIR31HG*) were upregulated in both groups (Figure 4).

#### LncRNAs, MIR31HG and LINC00939 Are Predicted to Interact with SP7 and DLX4 by RNA:RNA Base-Pairing

The RNA:RNA base-pairing interaction is an epigenetic mechanism played by lncRNAs to regulate the genic post-transcription, in order to repress or promote the transcripts translation process. Aiming to verify the possibility of RNA:RNA base-pairing interaction between the TFs *SP7* and *DLX4,* and the DElncRNAs that are upregulated in common among l-DMEM and l-OM groups used the prediction program RIblast, based on the LncRRIsearch webserver. There were predicted interactions between SP7:MIR31HG, SP7:LINC00939 (Figure 5), DLX4:MIR31HG, and DLX4:LINC00939 (Figure 6). All interactions presented negative minimum energy (MinEnergy) <12 kcal/mol, which characterizes a high affinity of binding between the transcripts. Both MIR31HG and LINC00939 were identified as 3′ polyadenylated (Appendix A). The results corroborate the probability of post-transcriptional and/or translational repressive regulation on *SP7* and *DLX4* by MIR31HG and LINC00939 base-pairing interactions and suggest that these interactions occur passively, without the need for enzymes or catalytic molecular agents, due to the negative binding energy values presented.

### 3.4. Machine Learning Program Predicted Potential for DNA:RNA Triplex-Forming for the SP7 DNA Site and for the lncRNA LINC00939

Another epigenetic mechanism mediated by lncRNAs is the DNA:RNA triplex-forming, which consists of interactions between lncRNAs and DNA-specific sequences, via triple-helix (triplex) formation, to inhibit or induce gene transcriptions. We performed prediction analysis of DNA:RNA triplex-forming potential for the *SP7* and *DLX4* DNA sites and for the MIR31HG and LINC00939 transcripts. Triplex-forming potential was predicted for the *SP7* DNA sites (score = 0.68) (Table 1) and for the first exon (*ENSE00002048191*) of one LINC00939 transcript (*ENST00000502479.1*) (score = 0.75) (Table 2). Conversely, there was no prediction of triplex-forming potential for the *DLX4* DNA site, nor for the lncRNA MIR31HG. These results point to a potential transcriptional regulation on the *SP7* DNA site (inhibition) by triplex formation with the lncRNA LINC00939.

### 3.5. MIR31HG and LINC00939 Transcripts Are Upregulated while SP7 and DLX4 Are Downregulated in l-PDLCs

In agreement with RNA-seq data, the expression of the lncRNAs, *MIR31HG* and *LINC00939,* were upregulated in l-PDLCs (× h-PDLCs) in both conditions, non-induced (DMEM) and induced (OM), with statistical significance for *MIR31HG* (l-DMEM × l-OM, *p* = 0.001; l-OM × h-OM, *p* = 0.001) and for *LINC00939* (l-DMEM × l-OM, *p* = 0.004; l-DMEM × h-DMEM, *p* = 00.15; l-OM × h-OM, *p* < 0.0001). In addition, the transcript levels of the osteogenic TFs *SP7* and *DLX4* in l-PDLCs were lower, with statistical significance for *SP7* (l-DMEM × l-OM, *p* = 0.002; l-DMEM × h-DMEM, *p* < 0.0001; h-DMEM × h-OM, *p* = 0.004; l-OM × h-OM, *p* < 0.0001), and for *DLX4* (h-DMEM × h-OM, *p* = 0.005; l-OM × h-OM, *p* < 0.0002) (Figure 7). These results show a negative correlation between *MIR31HG*/*LINC00939* and *SP7*/*DLX4* expression profiles, reinforcing the hypothesis of repressive action of these lncRNAs on the (post)transcriptional regulation of SP7 and DLX4.

Altogether, the results of chromatin accessibility, DNA methylation levels, RNA-seq, PCR, and predicted base-pairing interactions suggest *MIR31HG* and *LINC00939* might be good predictors of osteogenic commitment.

## 4. Discussion

The advancement of genomic sequencing techniques and bioinformatic tools has helped us to better understand the epigenetic machinery involved in the phenotype acquisition of MSCs. Chromatin conformation, DNA methylation pattern, transcriptional profile, and ncRNAs interactions are part of this machinery and impact cell differentiation potential. Here, we aimed to investigate the epigenetic machinery of the transcription factors (TFs), *SP7* and *DLX4*, involved in the osteoblastic differentiation, in PDLCs, using omic techniques and bioinformatic tools. We provided data that indicate *SP7* and *DLX4* might be regulated at transcriptional, post-transcriptional, and translational steps by the epigenetic mediators MIR31HG and LINC00939 and this could impact their osteogenic phenotype acquisition. So far, our results indicate *MIR31HG* and *LINC00939* might be good predictors of osteogenic commitment.

In our previous investigation, we showed an epigenetic landscape more accessible in l-PDLCs, when compared to h-PDLCs [12]; however, several TFs such as *DLX4* and *SP7* were pointed out to be downregulated in l-PDLCs. Here, in the ATAC-seq analyses, we found more accessible chromatin peaks for both, *SP7* and *DLX4* genomic regions in l-PDLCs, when compared to h-PDLCs, except to the transcription end site (TES) region of the *DLX4*. One of the hypotheses for this discrepancy would be the differential presence, concentration, and action of some epigenetic chromatin regulators, such as DNA methyltransferases (DNMTs) and histone deacetylases (HDACs), among the PDLCs populations. Nevertheless, to verify this hypothesis, it would be necessary to integrate other analyses that were not performed in this study, for example, chromatin immunoprecipitation (ChIP). The difference in peak score amplitude between l- and h-PDLCs was greater in *SP7* than in *DLX4*, and the peaks annotation also was distinct among the TFs genomic regions, with peaks concentration in exonic, TSS, and TES regions for *SP7*, and peaks concentration in intronic and TSS regions for *DLX4*. A study performed by Tai et al., in 2017, discovered a similar profile of chromatin accessibility on the *SP7* genomic region during the MC3T3 pre-osteoblast differentiation [42], with high chromatin accessibility centered within ~1 kb downstream of TES. Furthermore, this region coincided with CpG islands enrichments in *SP7*, indicating its importance in gene regulation. In the same, abovementioned study, accessible peaks were localized in the intronic region and in all osteoblastogenesis stages (pre-osteoblast, matrix deposition, nodule formation, and mineralization). These findings show chromatin conformation more favorable to *SP7* and *DLX4* transcription in l-PDLCs than h-PDLCs and suggest an important role of TES and intronic regions, i.e., non-promoter regions, on their gene regulations. 

The TFs also presented distinct DNA methylation patterns to each other, without differentially methylated probes (DMPs) between l- and h-PDLCs. The average methylation intensities (AMIs) remained below 0.5 in all groups, indicating an overall DNA methylation pattern tending to unmethylated, except in the 3′UTR region of the *DLX4*, which presented AMIs above 0.5, ranged from 0.7 to 0.8, indicating DNA methylation pattern tending to totally methylated in this region. Conversely, very low methylation intensities were found in the 3′UTR region of the *SP7*, ranging around 0.03, which matches the high chromatin accessibility presented in ATAC-seq results for this TES region. In 2019, Lhoumaud et al. examined the interplay between chromatin accessibility and DNA methylation in mouse embryonic stem cells, using the *EpiMethylTag* method, that combined ATAC-seq with bisulfite conversion and showed that DNA methylation rarely coexists with chromatin accessibility [43]. 

The signaling pathways are complex cascades that are dependent on a “perfect syntony” among many molecular events to promote gene transcriptions. In our RNA-seq analyses, we did not find differentially expressed genes (DEGs) amongst l- and h-PDLCs in the main signaling pathways involved in the TFs gene transcription, except for the *Sclerostin* (*SOST*), which was found to be downregulated in l-PDLCs. Recent studies show a correlation of expression between *SOST* and several osteoblastic TFs, such as *RUNX2* and *SP7* [44,45]. This protein plays an important role in the osteoblast development, acting as an inhibitor of the Wnt/βcatenin pathway by binding to the cell transmembrane receptors low-density lipoprotein receptor protein 5 and 6 (LRP5/6) and Frizzled. However, the other genes of this pathway were not found in DEGs. Several studies show the repressor effect of Sirtuin 1 (SIRT1) on the *SOST* expression of osteocytes [46,47]. In our RNA-seq dataset, the *SIRT1* was found to be slightly upregulated in l-DMEM (logFC = 0.17) and l-OM (logFC = 0.49), without significant difference. So far, the results suggest a transcriptional status of the signaling pathways more favorable to TFs’ expression in l-PDLCs compared to h-PDLCs.

Regarding the RNA-seq results for these TFs, our previous study [12] showed *SP7* and *DLX4* as DEGs and downregulated in both l-DMEM (x h-DMEM) and l-OM (x h-OM). However, these transcriptional profiles did not match the chromatin conformation or neither the DNA methylation pattern found here for these TFs. Both presented more accessible peaks, with higher peak scores in l-PDLCs than h-PDLCs, with DNA methylation patterns tending to be unmethylated in both populations. We expected to find opposite chromatin accessibility and DNA methylation patterns in l-PDLCs. Other studies also found discrepancies between ATAC-seq and RNA-seq results, and a positive correlation between DNA methylation and gene expression [48,49,50]. This paradox suggested the hypothesis of a fine-tuning gene regulation on *SP7* and *DLX4* in l-PDLCs, mediated by lncRNA interactions.

After we identified 18 lncRNAs DEGs in common between l-DMEM (x h-DMEM) and l-OM (x h-OM), only nine were upregulated in both groups, of which two (MIR31HG and LINC00939) were predicted by the web-based prediction tool to interact with *SP7* and *DLX4* by RNA:RNA base-pairing. This program is based on up-to-date benchmark data to compute the base-paring probabilities inter and intramolecular between RNA sequences [51]. 

The MIR31HG is a host gene of the MIR31, a miRNA validated as an SP7 repressor and osteogenesis inhibitor [52,53]. Nevertheless, to date, there are no studies correlating the *MIR31HG* and *SP7* expression profiles, nor predicting interactions between them. In 2018, Huang et al. cultivated BMSCs on titanium surfaces biofunctionalized with small interfering RNA (siRNA)-targeting MIR31HG and reported an increase in the relative expression of osteogenic genes such as *ALPL*, *RUNX2,* and *BGLAP*, without analyzing the *SP7* [54]. Since the MIR31HG hosts the MIR31, one possibility would be the MIR31HG produces precursors of MIR31 through intracellular shearing, and this represses the *SP7* [55]. However, the MIR31 was not found in DEG in our RNA-seq dataset, requiring a small RNA sequencing (miRNA-seq) to confirm this transcriptional profile. The LINC00939 is an Intergenic lncRNA, located on chromosome 12 (12q24.32), measuring 24,687 pairs of bases; although, we did not find previous publications about this lncRNA. Our hypothesis is both lncRNAs can regulate the post-transcriptional and/or translational processes of the *SP7* and *DLX4* by RNA:RNA base-pairing interactions. To the *SP7*, the interactions with MIR31HG were predicted in 5′UTR and coding sequence (CDS) regions, and LINC00939 was predicted with the majority being in the CDS region. In this case, both lncRNAs could repress the *SP7* post-transcription by “A to I” RNA edition, promoting hydrolytic deamination of adenosine to inosine [56], and/or the SP7 translation by the impediment of ribosome subunit binding. To the *DLX4*, the interaction with MIR31HG was predicted in the 5′UTR region, and LINC00939 predicted the majority in the 3′UTR region. In this case, the MIR31HG could repress the *DLX4* translation by the impediment of ribosome subunit binding, and the LINC00939 could repress the DLX4 post-transcription via Staufen-mediated decay, forming an intermolecular duplex with Alu element in the 3′UTR region [57,58]. Nonetheless, we should interpret these results with caution since the prediction tools “don’t consider complex structure folding as RNA tertiary structures, non-canonical base-pairing and co-transcriptional folding process and may contain false-positive predictions” [35]. Thus, experimental validations involving silencing and/or overexpression of these lncRNAs followed by SP7 and DLX4 protein analyses are necessary to authenticate the causality of these predicted interactions.

Since RNA:RNA base-pairing is a post-transcriptional and translational regulatory mechanism, we also investigated the probability of transcriptional regulation by DNA:RNA triplex forming. For this purpose, we use the TriplexFPP, a machine learning program “based on the experimentally verified data, where the high-level features are learned by the convolutional neural networks” [37], which increases the potential of a triplex formation in practice. In contrast, the scarce number of validation assays limits the program training, which greatly restricts the scope of predictions. In our analysis, it was predicted a triplex target site (TTS) for the SP7 gene and triplex-forming oligonucleotide (TFO) for the exon *ENSE00002048191* of the LINC00939 transcript *ENST00000502479.1*. We hypothesize that the SP7:LINC00939 triplex-forming represses the *SP7* transcription by steric hindrance of the RNA polymerase II at the promoter region [20,59]. However, the program does not predict matching between TTS–TFO, requiring experimental assays, such as chromatin oligo-affinity precipitation (ChOP), to confirm this interaction [60,61].

Finally, in an attempt to identify markers to assist with the selection of PDLCs with distinct osteogenic potential, we selected a panel of four genes for further confirmation by qPCR. The panel included both lncRNAs, *MIR31HG* and *LINC0093,9* and both osteogenic TFs, *SP7* and *DLX4*. The confirmation and combination of higher levels of *MIR31HG* and *LINC00939* transcripts alongside lower levels of *SP7* and *DLX4* gene expression in l-PDLCs after in vitro osteogenic induction indeed show potential in predicting distinct osteogenic potential amongst PDLCs. Surprisingly, there was no correlation between qPCR and RNA-seq results for the *DLX4* gene expression among the l- and h-DMEM. Everaert et al. also reported discrepancies among qPCR and RNA-seq gene expression measurements, typically in smaller genes [62]. Coincidentally, *DLX4* was the smallest gene analyzed here (sizes: *DLX4* ≈ 6 kb; SP7 ≈ 18 kb; LINC00939 ≈ 24 kb; MIR31HG ≈ 104 kb).

Further studies with functional approach by silencing and overexpressing the lncRNAs, *MIR31HG* and *LINC00939,* and the impact on *SP7* and *DLX4* gene expression will be necessary to confirm the hypothesis here identified.

## 5. Conclusions

The set of results obtained here indicates the lncRNAs, MIR31HG and LINC00939, as possible epigenetic mediators on PDLCs osteogenic phenotype commitment through repressive regulation of the osteogenic TFs, *SP7* and *DLX4* (Figure 8). In this regard, *MIR31HG* and *LINC00939* might be good predictors of osteogenic commitment in PDLCs.

## Figures and Tables

**Figure 1 genes-14-01649-f001:**
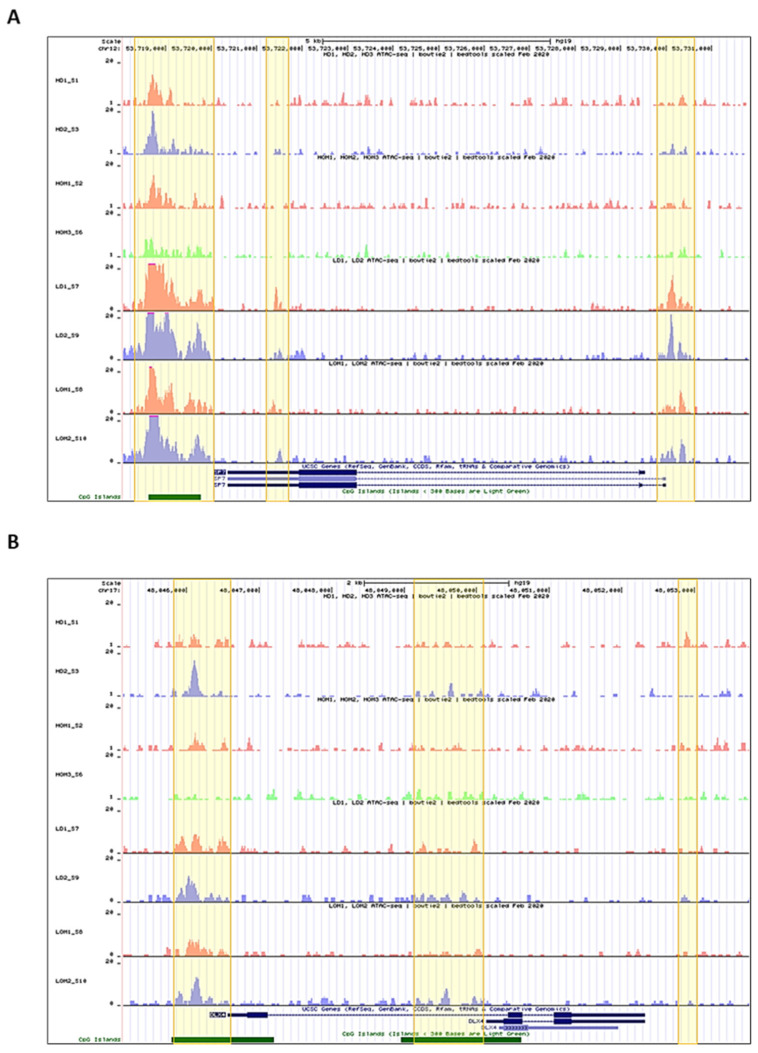
Overview of the chromatin accessibility on the *SP7* and *DLX4* gene regions in each h-PDLCs and l-PDLCs sample: The *bigwig* annotation tracks represent the accessible peaks on the *SP7* (**A**) and *DLX4* (**B**) gene regions in each h-PDLCs (DMEM and OM) and l-PDLCs (DMEM and OM) sample. Highlighted in yellow are the genomic regions with higher peak scores. *HD1_S1* and *HD2_S3* = h-DMEM samples; *HOM1_S2* and *HOM3_S6* = h-OM samples; *LD1_S7* and *LD2_S9* = l-DMEM samples; *LOM1_S8* and *LOM2_S20* = l-OM samples.

**Figure 2 genes-14-01649-f002:**
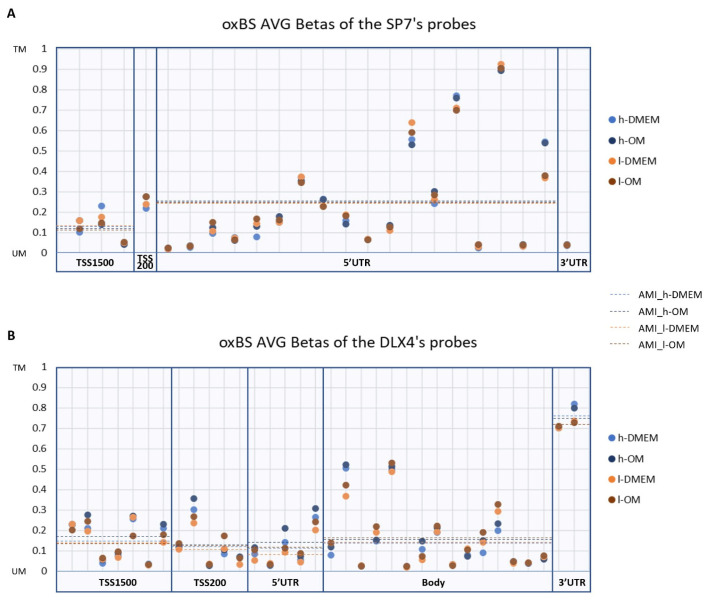
DNA methylation patterns in *SP7* and *DLX4* gene regions: The scatter plots represent the average β values (AVG-β) of each probe detected for *SP7* (**A**) and *DLX4* (**B**) gene regions in the oxidative bisulfite sequencing (oxBS) dataset. The dotted lines indicate the average methylation intensity (AMI) of the probes according to *UCSC_RefGene_Group*. TM: Totally Methylated; UM: Unmethylated.

**Figure 3 genes-14-01649-f003:**
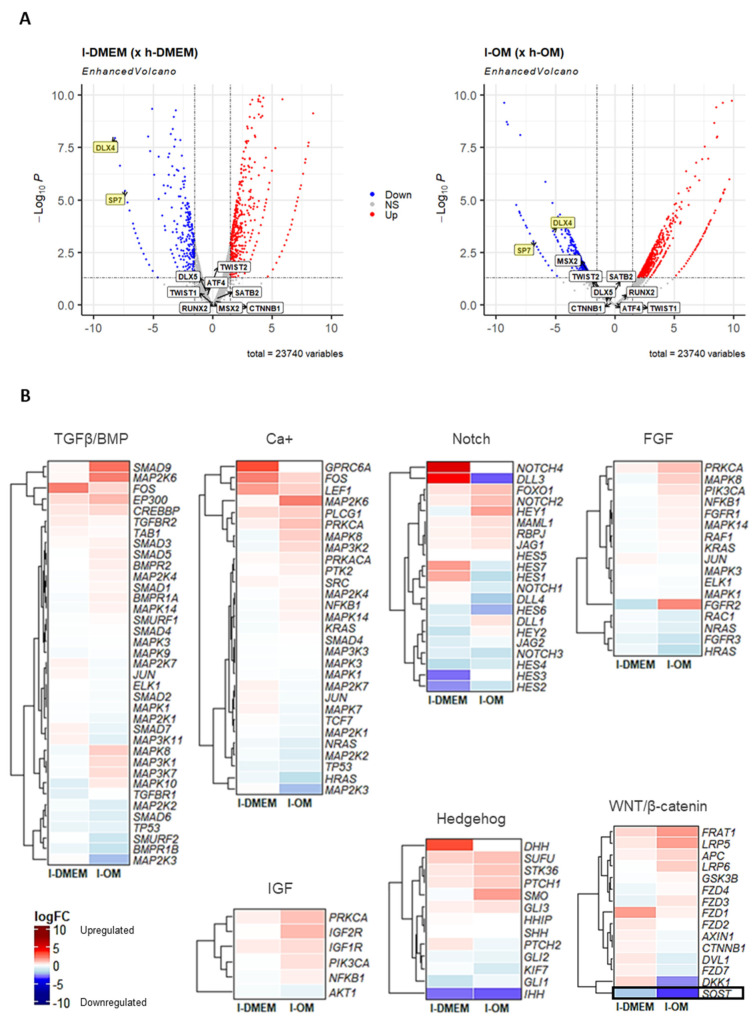
Analysis of other osteogenic TFs and signaling pathways in the RNA-seq dataset: The Volcano plots (**A**) show the magnitude of change in gene expression (logFC) and statistical significance (*p*-value) of the major osteogenic TFs, in l-DMEM (× h-DMEM) (left) and l-OM (× h-OM) (right). Highlighted in yellow are the *SP7* and *DLX4*, unique differentially expressed genes (logFC < −1.5 or >1.5 and *p*-value < 0.05) in both groups. Down: downregulated; Up: upregulated; NS: not significant. The Heatmap graphs (**B**) represent the logFC of the genes involved in the major osteogenic signaling pathways, in l-DMEM (× h-DMEM) and l-OM (× h-OM). It highlighted the *Sclerostin* (*SOST*) gene, a negative regulator of the WNT/β-catenin pathway, and was found to be differentially expressed (DEG) in both groups in l-PDLCs (× h-PDLCs). Reddish colors = upregulated; bluish colors = downregulated.

**Figure 4 genes-14-01649-f004:**
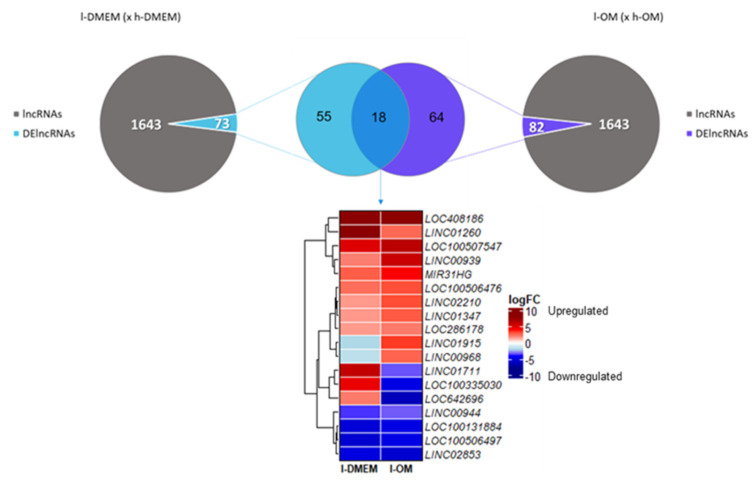
TFs Transcriptional Profiles: The Volcano plots show the magnitude of change in gene expression (logFC) and statistical significance (*p*-value) of the TFs, in l-DMEM (x h-DMEM) (**A**) and l-OM (x h-OM) (**B**). As highlighted, the SP7 and FABP4 genes are differentially expressed (logFC < −1.5 or >1.5 and *p*-value < 0.05). Down: downregulated; Up: upregulated; NS: not significant.

**Figure 5 genes-14-01649-f005:**
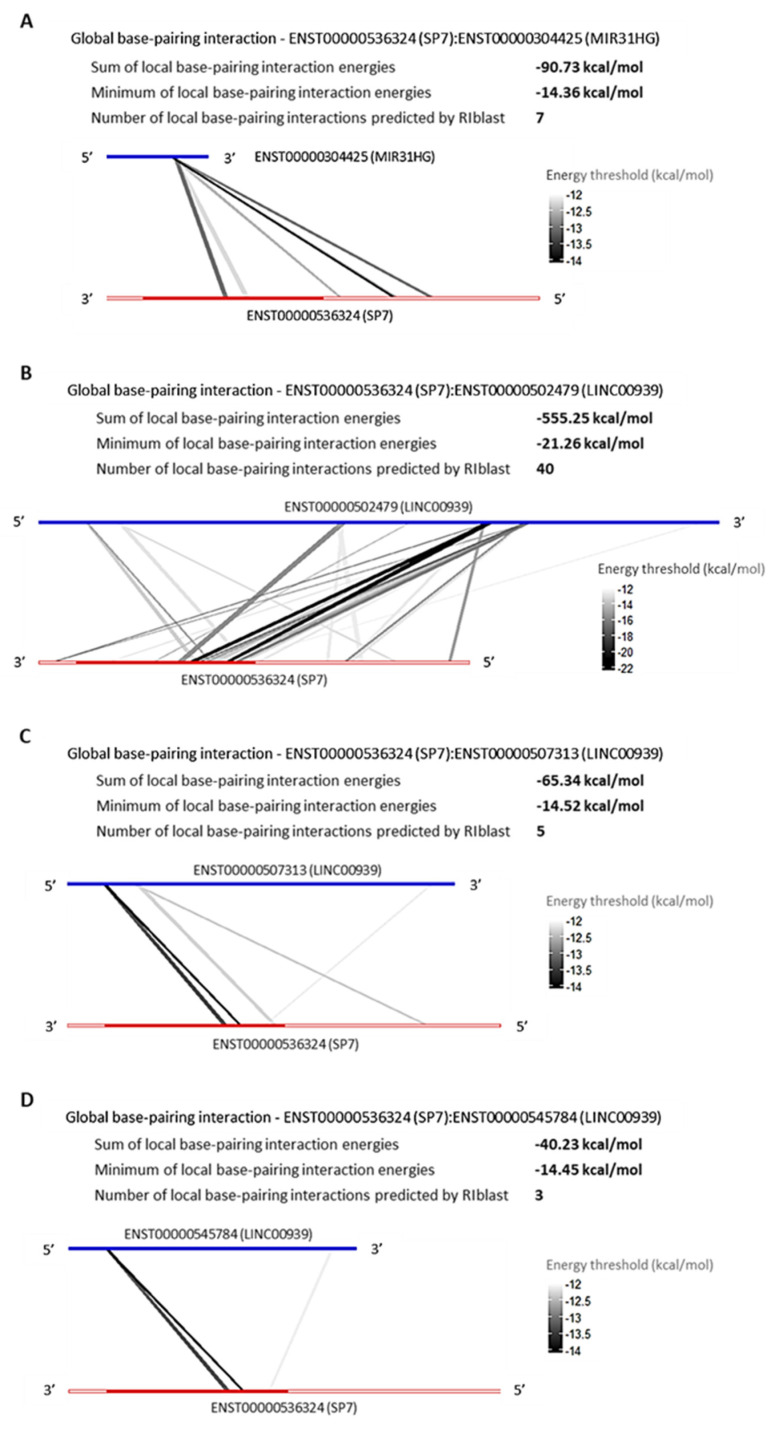
Global base-pairing interactions between SP7 and predicted DElncRNAs transcripts; Global base-pairing interactions between ENST00000536324 (SP7) and ENTS00000304425 (MIR31HG) (**A**), ENST00000502479 (LINC00939) (**B**), ENST00000507313 (LINC00939) (**C**), and ENST00000545784 (LINC00939) (**D**) transcripts. The blue bars represent the lncRNA transcripts and the red bars represent the SP7 transcript. The grayish lines represent the predicted interactions according to the energy threshold and SP7′s annotation (3′UTR, CDS, and 5′UTR regions).

**Figure 6 genes-14-01649-f006:**
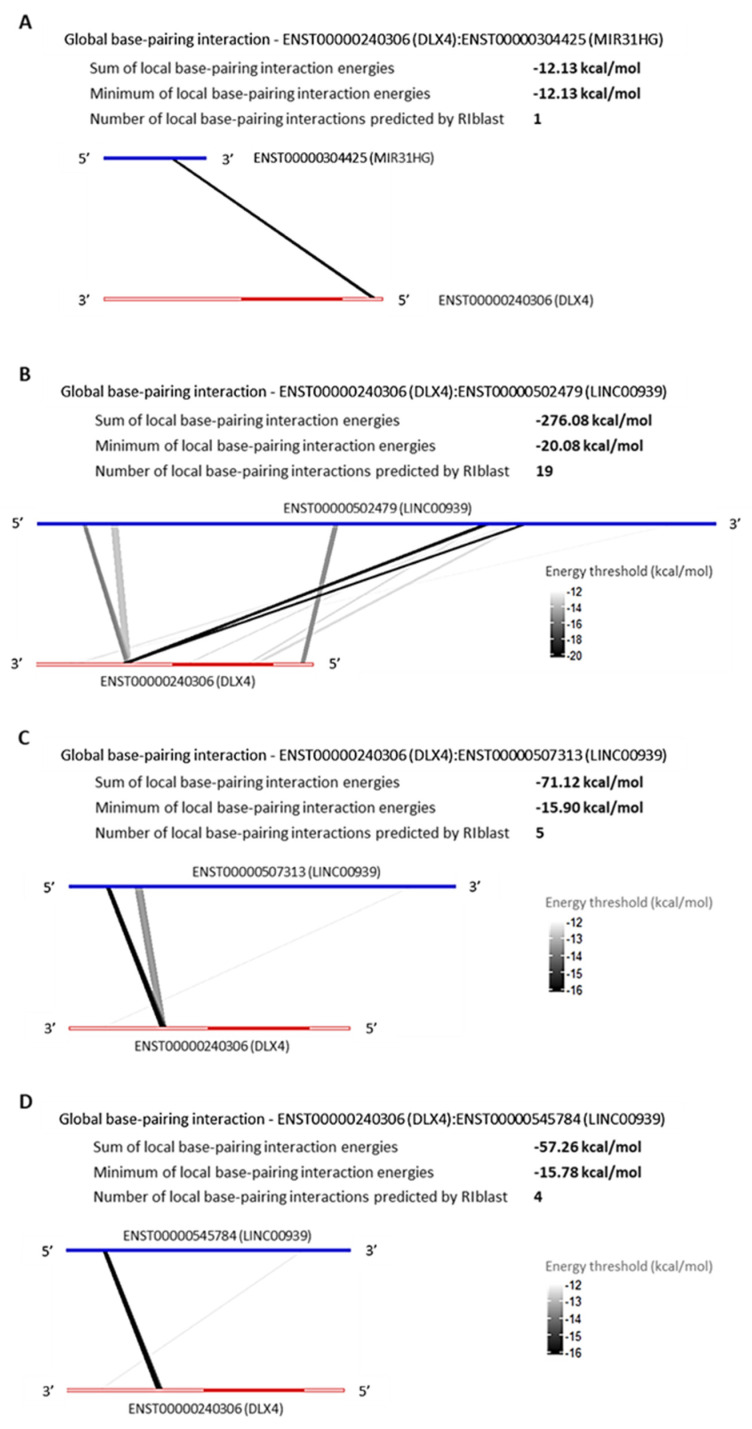
Global base-pairing interactions between DLX4 and predicted DElncRNAs transcripts; Global base-pairing interactions between ENST00000240306 (DLX4) and ENTS00000304425 (MIR31HG) (**A**), ENST00000502479 (LINC00939) (**B**), ENST00000507313 (LINC00939) (**C**), and ENST00000545784 (LINC00939) (**D**) transcripts. The blue bars represent the lncRNA transcripts and the red bars represent the DLX4 transcript. The grayish lines represent the predicted interactions according to the energy threshold and DLX4′s annotation (3′UTR, CDS, and 5′UTR regions).

**Figure 7 genes-14-01649-f007:**
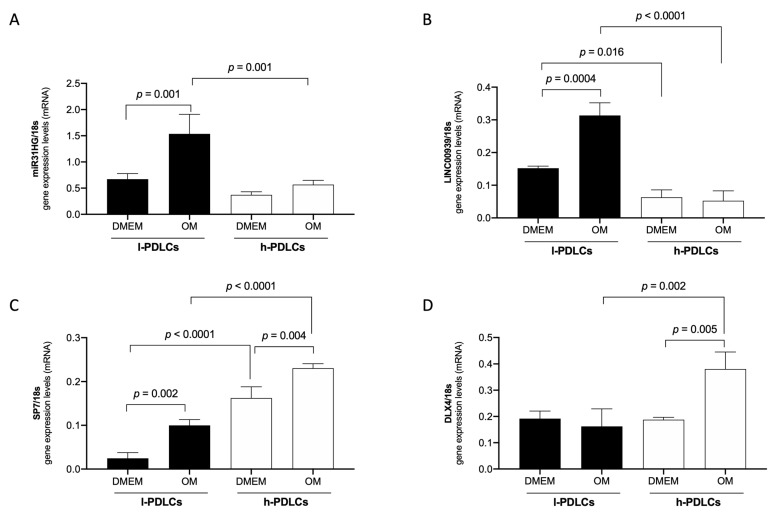
Gene expression levels of MIR31HG, LINC00939, SP7, and DLX4 in l- and h-PDLCs, cultured at 10 days in non-induced (DMEM) and osteogenic (OM) medium: qPCR analysis of MIR31HG (**A**), LINC00939 (**B**), SP7 (**C**), and DLX4 (**D**) in l- and h-PDLCs, cultured at 10 days in a non-induced (DMEM) and osteogenic (OM) medium. The results represent a mean ± standard deviation of three biological replicates, considering differential expression for *p* < 0.05.

**Figure 8 genes-14-01649-f008:**
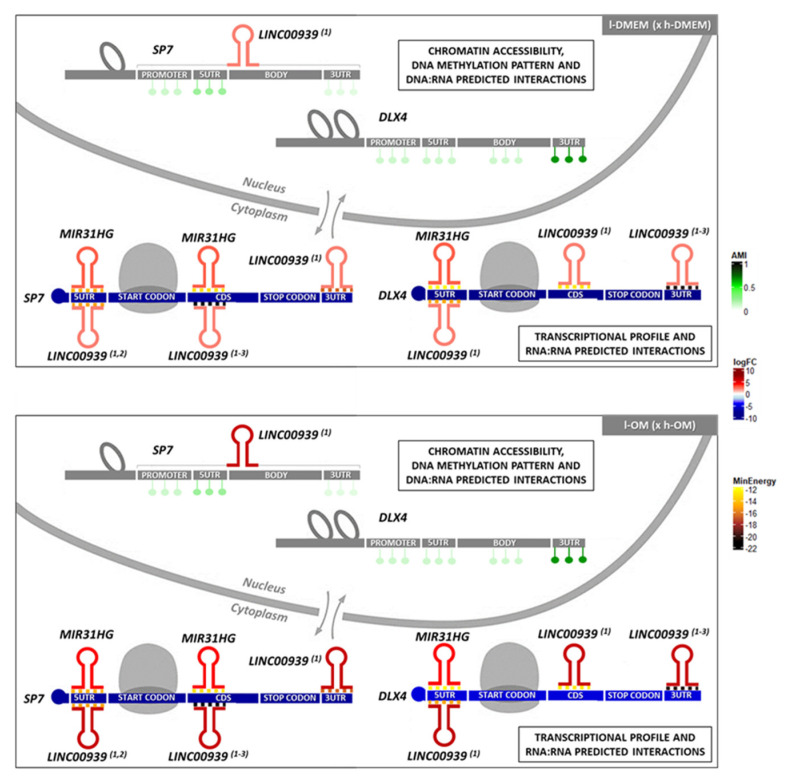
Schematic representation of the SP7 and DLX4 epigenetic status and their predicted interactions with the lncRNAs, MIR31HG and LINC00939, in l-PDLCs (× h-PDLCs) at 10 days, cultured in DMEM and OM medium: The results of the ATAC-seq dataset show the chromatin is more accessible for the SP7 and DLX4 genes in l-PDLCs (DMEM and OM) compared with h-PDLCs (DMEM and OM). The results of the Methylome dataset show a less methylated pattern for the SP7 and DLX4 genes, in all groups, except to the 3′UTR region of the DLX4, which exhibit a more methylated pattern. The lncRNAs analyses in the RNA-seq dataset show the DElncRNAs MIR31HG and LINC00939 (predicted to interact with SP7 and DLX4 by RNA:RNA base-pairing) upregulated in both l-DMEM (× h-DMEM) and l-OM (× h-OM) groups. The DNA:RNA interactions analyses predicted potential triplex-forming for one LINC00939 transcript (ENST00000502479) and for the SP7 DNA site, in practice. These data together allow us to hypothesize the existence of a possible transcriptional repressor regulation on the SP7 by triplex-forming with LINC00939, and a post-transcriptional repressor regulation on the SP7 and DLX4 transcripts by MIR31HG and LINC00939 base-pairing interactions, in l-PDLCs at 10 days, cultured in DMEM and OM medium. (1) = ENST00000502479; (2) = ENST00000507313; (3) = ENST00000545784; CDS = coding DNA sequence; reddish colors = upregulated; bluish colors = downregulated; greenish colors = average methylation intensity (AMI); yellowish and brownish colors = minimum energy (kcal/mol) of the predicted RNA:RNA interactions. Adapted from Gomes et al., 2020 [63].

**Table 1 genes-14-01649-t001:** Triplex prediction for the *SP7* and *DLX4* DNA sites.

TRIPLEX DNA SITE PREDICTION					
Gene Symbol	Assembly	Chromosome	Map Location	Score	Triplex-Forming
SP7	GRCh37	12	53720362:53739099	0.68235606	Triplex
DLX4	GRCh37	17	48046334:48052321	0.13426343	Nontriplex

**Table 2 genes-14-01649-t002:** Triplex-forming potential of the lncRNAs MIR31HG and LINC00939.

LNCRNAS TRIPLEX-FORMING POTENTIAL					
Gene Symbol	Transcript ID	Annotation	Exon ID	Score	Triplex-Forming
MIR31HG	ENST00000304425.3	exon	ENSE00001540342	0.0	Nontriplex
		exon	ENSE00001540341	0.0	Nontriplex
		exon	ENSE00001540339	0.0	Nontriplex
		exon	ENSE00001729409	0.0	Nontriplex
		intron 1		0.0	Nontriplex
		intron 2		0.0	Nontriplex
		intron 3		0.0	Nontriplex
LINC00939	ENST00000502479.1	exon	ENSE00002048191	0.7498492	Triplex
		exon	ENSE00002044944	0.0	Nontriplex
		exon	ENSE00001441774	0.0	Nontriplex
		exon	ENSE00002049964	0.033063784	Nontriplex
		exon	ENSE00002021220	0.30899337	Nontriplex
		exon	ENSE00002063907	0.0	Nontriplex
		intron 1		0.0	Nontriplex
		intron 2		0.0	Nontriplex
		intron 3		0.0	Nontriplex
		intron 4		0.0	Nontriplex
		intron 5		0.0	Nontriplex
LINC00939	ENST00000507313.1	exon	ENSE00002060110	0.0	Nontriplex
		exon	ENSE00002044944	0.0	Nontriplex
		exon	ENSE00001441774	0.0	Nontriplex
		exon	ENSE00002033127	0.0	Nontriplex
		exon	ENSE00002081221	0.0	Nontriplex
		exon	ENSE00002046518	0.0	Nontriplex
		intron 1		0.0	Nontriplex
		intron 2		0.0	Nontriplex
		intron 3		0.0	Nontriplex
		intron 4		0.124810636	Nontriplex
		intron 5		0.0	Nontriplex
LINC00939	ENST00000545784.1	exon	ENSE00002206498	0.0	Nontriplex
		exon	ENSE00002044944	0.0	Nontriplex
		exon	ENSE00002228181	0.0	Nontriplex
		exon	ENSE00002271284	0.0	Nontriplex
		exon	ENSE00002021220	0.30899337	Nontriplex
		exon	ENSE00002216415	0.0	Nontriplex
		intron 1		0.0	Nontriplex
		intron 2		0.0	Nontriplex
		intron 3		0.124810636	Nontriplex
		intron 4		0.0	Nontriplex
		intron 5		0.0	Nontriplex

## Data Availability

The data presented in this study are available on request from the corresponding authors. The data are not publicly available due to potentially identifiable genomic information.

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
