# Peer review of "Analyzes In Silico Indicate the lncRNAs MIR31HG and LINC00939 as Possible Epigenetic Inhibitors of the Osteogenic Differentiation in PDLCs"

_genes, 2023, doi:10.3390/genes14081649_

Round 1
Reviewer 1 Report
The title of manuscript is remarkable. English language has good quality. Figures have acceptable quality. There are some explainations that are needed about materials and methods and also Discussions.
1. About the section "Materials and Methods" in page 2
+ why you have not briefly explain about the culture of PDLCs?
+ Please explain how you identified PDLCs?
+ The authors should explain that how mineral deposition has affected the classification of PDLCs into two subgroup of l-PDLCs and h-PDLCs?
+ Please explaim briefly about "The oxidative bisulfite conversion reaction" in the main text of the manuscript which you have mentioned
in line 139-140
2. In page 9 line 298-299
The authors have mentioned "suggests these interactions occur spontaneously."
Please explain what do you mean by spontaneously?
3. About line 339-341
Why the authors have mentioned this interpretation? Please explain in detail.
4. About the section "Discussion"
Why you have not discussed about the future, obstacles and insights of the field of present study? Specially in future clinilac trials
5. Please check and adjust the "Reference list" based on the regulations of reference list of journal. (Titles, doi, the name of journal
and ... )
Author Response
REVIEWER 1.
The title of manuscript is remarkable. English language has good quality. Figures have acceptable quality. There are some explanations that are needed about materials and methods and also Discussions.
- About the section "Materials and Methods" in page 2
+ why you have not briefly explain about the culture of PDLCs?
+ Please explain how you identified PDLCs?
+ The authors should explain that how mineral deposition has affected the classification of PDLCs into two subgroups of l-PDLCs and h-PDLCs?
Thank you for pointing this out; we have now clarified our approach, which is highlighted in yellow in the revised version of the manuscript (Material and methods – Cell acquisition and culture). In brief, after the cells were collected from the donors, they were induced towards osteogenic differentiation and characterized by flow cytometry, population doubling time, alizarin red and gene expression (Assis et al. 2021. DNA and Cell Biology. doi: 10.1089/dna.2020.6239 and Assis et al. 2022. Cells. doi: 10.3390/cells11071126) (refs. 10,12). Based on this, the two PDLCs populations were classified into high and low osteogenic potential, i.e., the population from one donor showed higher capacity to produce mineral nodules in vitro, when compared to the PDLCs population from the second donor. Next, we performed three independent biological experiments involving 10 days of osteogenic stimulation for each one of the PDLCs (three for h-PDLCs and three for l-PDLCs). The obtained samples were then processed to perform the DNA (hydroxy)methylome, the ATAC-seq and the RNA-seq.
+ Please explain briefly about "The oxidative bisulfite conversion reaction" in the main text of the manuscript which you have mentioned in line 139-140.
Thank you for giving us the opportunity to explain. Since bisulfite does not distinguish between 5mC and 5hmC, new approaches have been proposed, such as Infinium MethylationEPIC BeadChip array (Illumina) coupled with the TrueMethyl™ Kit (Cambridge Epigenetix-CEGX, Cambridge, UK). The traditional bisulfite conversion leads to potential overestimation of 5mC levels due to the presence of 5hmC, the oxBS method generates a more accurate 5mC profile by removing the confounding factor of 5hmC, and both chemical modifications can be differentially identified by comparing methylated CpG sites between the BS- and oxBS-treated samples.
The workflow before the (Hydroxy)methylation array using the MethylationEPIC BeadChip (Illumina) comprises two main steps, as follow:
- Oxidative bisulfite (oxBS) treatment: The same sample (1 μg) is splitted in two equal reactions (500 ng each), one of which undergoes chemical oxidation followed by bisulfite conversion and the other undergoes mock oxidation (oxidant replaced by water) followed by bisulfite conversion. The oxidant solution converts 5hmC bases to its formyl derivative 5fC, which is deaminated to uracil by bisulfite treatment; therefore, by selectively oxidizing 5hmC to 5fC prior to bisulfite treatment, only 5mC remains unconverted by bisulfite treatment.
- Bisulfite conversion: In the presence of bisulfite, 5fC is deformylated and deaminated to uracil, after the following steps: bisulfite-mediated conversion of unmethylated cytosines and formylated cytosines to uracil, cleanup of bisulfite converted DNA with magnetic beads, and desulfonation and elution of DNA.
The Infinium Methylation EPIC BeadChip enables to study DNA (hydroxy)methylation quantitatively at 830,000 sites, including CpG sites, RefSeq genes, ENCODE open chromatin, ENCODE transcription factor binding sites, and FANTOM5 enhancers. Since enhancers possess the highest enrichment of 5hmC among all genomic elements, a substantial proportion of potentially 5hmC-dependent functional genomic elements are detected by the Infinium Methylation EPIC array.
- In page 9 line 298-299
The authors have mentioned "suggests these interactions occur spontaneously."
Please explain what do you mean by spontaneously?
Thank you for giving us the opportunity to clarify this. The spontaneity of these interactions is suggested due to the negative binding energy values presented, which characterizes passive molecular interactions, without the need for catalytic agents. We changed the pointed mention for "suggests that these interactions occur passively, without the need for enzymes or catalytic molecular agents, due to the negative binding energy values presented."
- About line 339-341
Why the authors have mentioned this interpretation? Please explain in detail.
Thank you for the opportunity to better explain this. In fact, since this is a manuscript with several and consecutive steps of analysis, we have added some partial conclusions every step of the way, in general, at the end of each key points of the manuscript. We think this could be good to lead the reasoning of the readers to this final point, which is the very core of the manuscript conclusion itself. Sometimes, there are readers that are not so familiarized with this type of manuscript, containing more in deep Bioinformatic analysis. In this case, partial conclusions from time to time, can be a good way to make easier to the readers. To better understand this flow, we added a specific partial conclusion about PCR results (“These results show a negative correlation between MIR31HG/LINC00939 and SP7/DLX4 expression profiles, reinforcing the hypothesis of repressive action of these lncRNAs on the (post)transcriptional regulation of SP7 and DLX4”) before of this mentioned interpretation.
- About the section "Discussion"
Why you have not discussed about the future, obstacles and insights of the field of present study? Specially in future clinical trials.
Thank you for pointing this out. Some methodological obstacles were pointed out in the Discussion, such as the absence of chromatin immunoprecipitation (ChIP) and chromatin oligo-affinity precipitation (ChOP) assays for the differential analysis of chromatin regulatory epigenetic agents (HDACs and DNMTs) among PDLC populations (line 364-366) and confirmation of the SP7:LINC00939 triplex-formation, respectively (line 465-467). In addition, limitations of the prediction tools used also were pointed in the Discussion (line 449-451 and 460-461). Future clinical trials were not discussed here because this study is based almost exclusively on in silico analyses, being necessary firstly to carry out laboratory experimental assays of silencing and overexpression of these target genes to confirm the causality of the predicted interactions between them (line 451-454 and 480-482).
- Please check and adjust the "Reference list" based on the regulations of reference list of journal (Titles, doi, the name of journal and ...).
We apologize for that. The reference list was adjusted according to the journal guideline.
Reviewer 2 Report
In this manuscript, the authors performed ATAC-seq, methylome and RNA-seq in human periodontal ligament mesenchymal cells (PDLCs) that were cultured in non-induced and osteogenic medium for 10 days. They predicted that the lncRNAs MIR31HG and LINC00939 might inhibit the osteogenic differentiation in PDLCs by transcriptional and/or translational repression of the transcription factors SP7 and DLX4. Although the datasets in this study provide useful resources for further exploration of osteogenic differentiation. I have the following concerns:
1. If possible, it will be great to provide evidence regarding the osteogenic potentials of l-PDLCs and h-PDLCs that were sampled for bioinformatics.
2. The authors stated that ‘Three independent experiments were performed for each PDLCs, with three technical replicates for each one, with cells in the passages P5-P8, except when stated otherwise’. However, this is impossible to judge from the presented data. It is important to illustrate how the data look, and how close the replicates are.
3. Besides SP7 and DLX4, I or probably most readers would like to know whether there are any other transcription factors potentially involved in osteogenic differentiation in this study.
4. It is disappointing in general that no experimental validations of predicted interactions between MIR31HG/LINC00939 and SP7/DLX4 in the paper.
Author Response
REVIEWER 2.
In this manuscript, the authors performed ATAC-seq, methylome and RNA-seq in human periodontal ligament mesenchymal cells (PDLCs) that were cultured in non-induced and osteogenic medium for 10 days. They predicted that the lncRNAs MIR31HG and LINC00939 might inhibit the osteogenic differentiation in PDLCs by transcriptional and/or translational repression of the transcription factors SP7 and DLX4. Although the datasets in this study provide useful resources for further exploration of osteogenic differentiation. I have the following concerns:
- If possible, it will be great to provide evidence regarding the osteogenic potentials of l-PDLCs and h-PDLCs that were sampled for bioinformatics.
Thank you for pointing this out; we have now clarified our approach, which is highlighted in yellow in the revised version of the manuscript (Material and methods – Cell acquisition and culture). In brief, after the cells were collected from the donors, they were induced towards osteogenic differentiation and characterized by flow cytometry, population doubling time, alizarin red and gene expression (Assis et al. 2021. DNA and Cell Biology. doi: 10.1089/dna.2020.6239 and Assis et al. 2022. Cells. doi: 10.3390/cells11071126) (refs. 10,12). Based on this, the two PDLCs populations were classified into high and low osteogenic potential, i.e., the population from one donor showed higher capacity to produce mineral nodules in vitro, when compared to the PDLCs population from the second donor.
- The authors stated that ‘Three independent experiments were performed for each PDLCs, with three technical replicates for each one, with cells in the passages P5-P8, except when stated otherwise’. However, this is impossible to judge from the presented data. It is important to illustrate how the data look, and how close the replicates are.
Thank you for your concern, we appreciate the chance to better explain. Figure 1, which is the overview of the chromatin accessibility on the SP7 and DLX4 gene regions in each h-PDLCs and l-PDLCs sample is one good example of how close the replicates are. In this panel, the heigh of peaks can be observed amongst the samples, individually, and, therefore, it is possible to notice the homogeneity between the independent experiments carried out. Another very good example is the Figure 7, which shows the gene expression levels of MIR31HG, LINC00939, SP7 and DLX4 in l- and h-PDLCs, cultured at 10 days in non-induced (DMEM) and osteogenic (OM) medium. The PCR runs results are illustrated through bar graphics, alongside with mean and standard deviations. As can be observed, the standard deviations are small and, therefore, show the homogeneity amongst experimental repetitions. Another relevant detail is that the normality test was applied to observe the distribution of the data. After the normal distribution was confirmed, we have performed a parametric test. All these facts illustrate how close the replicates are.
To make even more clear, we have added a paragraph to the manuscript to explain how the PCR statistics were performed and it is highlighted in yellow in the manuscript. Please, see below:
STATISTICAL ANALYSIS OF THE PCR DATA
Data were initially examined for normality by Shapiro-Wilk test and expressed as mean ± standard deviation (SD). After normal data distribution was confirmed, One-way analysis of variance (ANOVA α £ 0.05) followed by pairwise multiple-comparison test (Tukey) were used to identify the difference amongst groups (GraphPad Prism 7 – GraphPad Software Inc., San Diego, CA, USA).
- Besides SP7 and DLX4, I or probably most readers would like to know whether there are any other transcription factors potentially involved in osteogenic differentiation in this study.
Thank you for your observation. In this study, we analysed only the epigenetic overview of the osteogenic transcription factors SP7 and DLX4, identified as DEGs in a previous study by our group. Other osteogenic transcription factors, such as RUNX2, CTNNB1, SATB2, ATF4, MSX2, DLX5 and TWIST1, were not found DEGs in the RNA-seq dataset. However, by way of clarification for the readers, we have added the analysis of the transcriptional profile of these other osteogenic TFs.
- It is disappointing in general that no experimental validations of predicted interactions between MIR31HG/LINC00939 and SP7/DLX4 in the paper.
Thank you for bringing this up; we agree with the reviewer, however, it is important to highlight this paper is of novelty and brings very new information. Of course, it is a first step, and we are planning further and more in deep validation in the future, including the increase in the number of PDLCs presenting distinct osteogenic potential and other methodologies such as silencing the miR31HG by CRISPR technology. After the silencing of miR31HG in the next few months, we will carry on several experimental validations, including the cells osteogenic response with the miR31HG silenced or not in culture cells, titanium surface and animal studies. However, this could be taking years to have a final conclusion about these relevant questions first raised here.
Round 2
Reviewer 1 Report
I dont have more suggestion.
Reviewer 2 Report
NA.